# Novel fold of rotavirus glycan-binding domain predicted by AlphaFold2 and determined by X-ray crystallography

Liya Hu [1✉], Wilhelm Salmen[1], Banumathi Sankaran [2], Yi Lasanajak[3], David F. Smith[3], Sue E. Crawford[4], Mary K. Estes [4,5] & B. V. Venkataram Prasad [1,4✉]

The VP8* domain of spike protein VP4 in group A and C rotaviruses, which cause epidemic gastroenteritis in children, exhibits a conserved galectin-like fold for recognizing glycans during cell entry. In group B rotavirus, which causes significant diarrheal outbreaks in adults, the VP8* domain (VP8*B) surprisingly lacks sequence similarity with VP8* of group A or group C rotavirus. Here, by using the recently developed AlphaFold2 for ab initio structure prediction and validating the predicted model by determining a 1.3-Å crystal structure, we show that VP8*B exhibits a novel fold distinct from the galectin fold. This fold with a β-sheet clasping an α-helix represents a new fold for glycan recognition based on glycan array screening, which shows that VP8*B recognizes glycans containing N-acetyllactosamine moiety. Although uncommon, our study illustrates how evolution can incorporate structurally distinct folds with similar functionality in a homologous protein within the same virus genus.

[1] Verna and Marrs McLean Department of Biochemistry and Molecular Biology, Baylor College of Medicine, Houston, TX, USA. [2] Berkeley Center for Structural Biology, Molecular Biophysics and Integrated Bioimaging, Lawrence Berkeley Laboratory, Berkeley, CA, USA. [3] Emory Glycomics and Molecular Interactions Core (EGMIC), Emory University School of Medicine, Atlanta, GA, USA. [4] Department of Molecular Virology and Microbiology, Baylor College of Medicine, Houston, TX, USA. [5] Department of Medicine, Baylor College of Medicine, Houston, TX, USA. ✉email: lhu@bcm.edu; vprasad@bcm.edu

Rotaviruses are non-enveloped icosahedral double-stranded RNA (dsRNA) viruses belonging to the *Reoviridae* family[1]. These viruses exhibit enormous genetic and serological diversity. Based on the sequence and antigenic differences of the capsid protein VP6, they are classified into ten different species or groups (A–J)[2,3]. Group A, B, C, and H rotaviruses infect both humans and animals[4,5]. Epidemiologically, groups A, B, and C are the best characterized. While group A rotaviruses (RVA) and to a lesser extent group C rotaviruses (RVC) are the causative agents of most gastroenteric infections worldwide, the group B rotaviruses (RVB) have been associated with large epidemic outbreaks of severe gastroenteritis in China[6,7] and sporadic infection in several countries[8–10]. Unlike RVA, which infects mainly young children, RVB causes cholera-like severe diarrhea predominantly in adults, although children can be infected as well[8]. Antibodies to the RVB have been detected in people in developed countries such as the USA, Canada, and the UK[11,12], indicating a broader prevalence of RVB. RVB also infects animals and caused recent hemorrhagic diarrheal outbreaks in piglets and foals, resulting in significant economic impact and posing a threat of potential zoonotic transmission[13,14]. Considering that these rotaviruses have segmented dsRNA genomes, with a propensity for evolving by gene reassortment from co-infections and mutations, the potential for the emergence of new variants that can cause severe epidemics cannot be discounted. A case in point is the current ongoing global COVID-19 pandemic caused by SARS-CoV-2[15].

Like RVA, which remains the best characterized thus far, and RVC, the genome of RVB consists of 11 dsRNA segments that encode 11 proteins[16]. Cryo-EM reconstructions of RVA have shown that trimers of VP4 form 60 protruding spikes attached to the outer VP7 and middle VP6 layers of the triple-layered capsid[17–19]. The proteolytic treatment of VP4, which significantly enhances infectivity, results in two fragments, VP8* and VP5*, that remain associated with the virion. Sequence comparison of the structural proteins encoded by different rotavirus groups shows that the VP8* domain of the spike protein VP4 is the most variable[20]. Extensive structural studies have shown that the galectin-like VP8* of human RVA and RVC (VP8*A and VP8*C) recognize various cellular glycans in a genotype-dependent manner (Fig. 1a, b)[21–29]. The VP8* of human RVA exhibits genotype-dependent glycan specificity by recognizing different histo-blood group antigens (HBGA)[30]. Differential recognition of HBGA has provided a possible rationale for why some RVAs specifically infect neonates, and some cause sporadic outbreaks while others infect a wider population[22–24,31]. Unlike the VP8*A and VP8*C, the structure of VP8*B and its glycan specificity have not been characterized. The VP8*B shares no sequence identity with either VP8*A or VP8*C (Supplementary Fig. 1a and Supplementary Table 1), which could differentially impact not only the structure but also the glycan-binding properties. Here we show by determining the crystal structure of VP8*B, using a model predicted using the recently developed AlphaFold2, that the VP8*B exhibits a fold with a twisted β-sheet clasping an α-helix that is entirely different from VP8*A or VP8*C. Our glycan array screening and in silico docking analysis show VP8*B recognizes glycans containing N-acetyllactosamine exemplifying how viruses can evolve by incorporating structurally distinct modules with similar functionality.

## Results

### The AlphaFold2 model of VP8*B reveals a novel fold. To understand the structure and the glycan specificity of VP8*B, we undertook crystallographic studies of VP8*B and performed glycan array screening. As a representative VP8*B, the amino acid sequence of a human RVB, isolated from a gastroenteritis outbreak in India[32], was expressed and purified for structure determination by X-ray crystallography. The VP8*B crystals diffracted to 1.3-Å resolution. Consistent with the low sequence identity with either VP8*A or VP8*C (Supplementary Fig. 1a and Supplementary Table 1), our attempts to find a molecular replacement (MR) solution using their structures or the structural models predicted from computational programs, such as trRosetta[33] and I-TASSER[34], was unsuccessful. Instead of using the traditional single anomalous dispersion (SAD) phasing method with crystallization of the selenium-methionine substituted recombinant VP8*B, we used the recently developed AlphaFold2 to generate a suitable search model[35,36].

The predicted AlphaFold2 models were significantly different from any of the previously determined experimental structures of RV VP8*, potentially representing a divergent novel fold for a glycan-binding protein (Fig. 1c). In the AlphaFold2 model, the residues 83–192 fold into several β-strands forming a twisted β-sheet clasping a central α-helix, while the N- and C-terminal residues (aa 65–82 and aa 193–233) projecting away from this fold are flexible. To examine if a similar novel fold is also predicted for the VP8*B of other RVBs, we used the primary sequences of murine, bovine, and porcine RVB (Fig. S1a). The predicted fold for these sequences, despite only ~27–33% sequence identity, is the same as that predicted for VP8*B of human RVB (Supplementary Fig. 1b and Supplementary Table 1), suggesting that group B VP8* has diverged significantly from the well-characterized galectin-like fold of the VP8*A and VP8*C.

### Crystal structure of VP8*B. We used the well-ordered region of VP8*B (residues 83–192) in the AlphaFold2-predicted fold as a search model for MR, and determined the crystal structure of native VP8*B at 1.3-Å resolution (Table 1). The crystallographic asymmetric unit contains two VP8*B molecules in addition to a short peptide (Fig. 2a). These two molecules in the asymmetric unit of the crystal structure superimposed with a Cα RMSD of 0.037 Å using Secondary Structure Matching (SSM) superpose in COOT[37]. Although we purified and crystallized the recombinant VP8*B containing residues 65–233 of VP4, only the residues F78-N202 are observed in the crystal structure. The electron density of the N- and C-terminal residues are not observed due to their flexibility or a proteolytic cleavage that may have occurred during crystallization. The small peptide with four residues could be the protease cleavage product co-crystallized with the globular portion of the VP8*B. The electron density of the side chains does not match any residues of the N or C terminus of VP8*B, possibly due to heterogeneity of the short peptide and the lack of sequence specificity of the peptide-binding (Supplementary Fig. 2). The VP8*B structure contains seven β-strands and one α-helix (Fig. 2b, c). Five of these β-strands form a twisted antiparallel β-sheet that surrounds a central α-helix. In the primary sequence, this central α-helix is between the residues that form β4 and β5 strands.

To examine if this fold is unique, we used DALI server[38] to compare with other structures in the Protein Data Bank (PDB). In this comparison, no structures showed strong similarities. The two structures that showed a marginal similarity, with Z-scores of 3.6 and 3.5, respectively, are exo-inulinase and bacteriophage T4 gene product 9 (gp9) (Fig. 2d, e). Although these structures showed a similar disposition of the antiparallel β-sheet, they both lacked the central α-helix in the VP8*B structure. The RMSD between the matching 66 Cα atom pairs in the β-sheet region of the exo-inulinase is 2.7 Å, whereas the RMSD between 76 Cα atom pairs with gp9 is 3.6 Å (Fig. 2d, e).

Surprisingly, this DALI search did not identify similarity with either VP8*A or VP8*C structures deposited in the PDB,

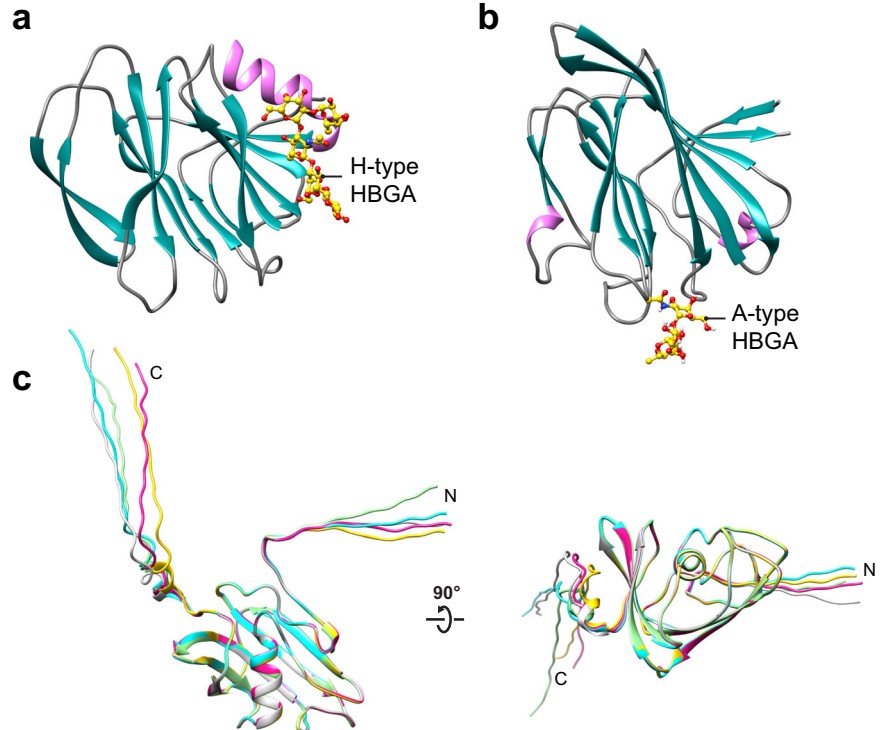

**Fig. 1 Ab initio modeling a human group B rotavirus VP8* with AlphaFold2 reveals a novel fold. a, b** Representative VP8* crystal structures: VP8*A in complex with H-type HBGA pentasaccharide (PDB ID: 5VX5) and VP8*C in complex with A-type HBGA trisaccharide (PDB ID: 5ZHO). The β-sheets and α-helixes are colored in sea green and pink, respectively. **c** Top 5 ranked predicted models of VP8*B shown as ribbon representations in different colors.

**Table 1 Data collection and refinement statistics.**

**RVB VP8* (PDB ID: 7RSW)**

| | |
|---|---|
| *Data collection* | |
| Space group | P 1 2₁ 1 |
| Cell dimensions | |
| *a, b, c* (Å) | 31.277, 116.257, 31.312 |
| *α, β, γ* (°) | 90, 103.54, 90 |
| Resolution (Å) | 50.0–1.32 (1.34–1.32) |
| $R_{merge}$ | 0.04 (0.262) |
| CC1/2 | 0.995 (0.844) |
| *I/σI* | 19.03 (2.31) |
| Wilson B-factor | 13.24 |
| Completeness (%) | 89.4 (61.4) |
| Redundancy | 1.9 (1.4) |
| *Refinement* | |
| Resolution (Å) | 30.41–1.32 (1.36–1.32) |
| No. reflections | 41,757 (1675) |
| $R_{work}/R_{free}$ (%) | 15.74 (22.41)/20.84 (34.18) |
| Number of atoms | |
| Protein | 1,914 |
| Water | 285 |
| *B*-factors | |
| Protein | 16.99 |
| Water | 34.01 |
| R.M.S.deviations | |
| Bond lengths (Å) | 0.005 |
| Bond angles (°) | 0.84 |
| Ramachandran favored (%) | 98.79 |
| Ramachandran allowed (%) | 1.21 |
| Ramachandran outlier (%) | 0 |

Statistics for the highest-resolution shell are shown in parentheses.

although the galectin-like fold in these structures exhibits twisted antiparallel β-sheet domains. We then examined if the β-sheet in the VP8*B could be aligned with either of the two β-sheets in the galectin-like fold of VP8*A or VP8*C structures. We found that the β-sheet structure of VP8*B is indeed unique, and shared no similarity with the β-sheets in VP8*A or VP8*C structures indicating the evolutionary path of VP8*B fold is distinct.

**Comparison of the experimental and AlphaFold2 models of VP8*B.** Having determined the structure of the VP8*B, we examined how closely the experimentally determined structure of VP8*B compared with the AlphaFold2-predicted model. Structural comparison of the two shows a high degree of similarity between the models with a Cα RMSD of 0.398 Å and 0.403 Å with the two VP8*B molecules in the asymmetric unit, respectively (Fig. 3a, b). Further inspection showed that the sidechain orientations also match well except for His131 (Fig. 3c, d). The sidechain orientation of this residue is likely influenced by the intermolecular contacts and solvent in the crystal. In the experimental structure, the side chain of His131 hydrogen bonds with Gln195 within the same molecule and Glu120 of the neighboring symmetry-related molecule (Fig. 3d). There is also a water molecule that hydrogen bonds with His131 sidechain and the main chain carbonyl oxygen atom of Cys198. In the Alpha-Fold2 model, the side chain of His131 residue orients differently, and the same orientation would clash with the water molecule observed in the crystal structure.

**AlphaFold model of full-length VP4B.** The accuracy of the AlphaFold2-predicted fold of VP8*B prompted us to investigate other regions in the VP4 spike of RVB, particularly in comparison with the VP4 structure of RVA (Supplementary Fig. 3a). The structural organization of the VP4 spike in the RVA virion is well characterized by cryo-EM studies[17,18]. In addition to the VP8* domain, the remainder of the spike protein VP4 in RVA consisting of VP5* is described as having a central body and a foot region that is buried inside the VP7 and VP6 layers at one of the channels of the icosahedral capsid (Supplementary Fig. 3b). The residues (248–479) that form the central body of the spike fold into a β-

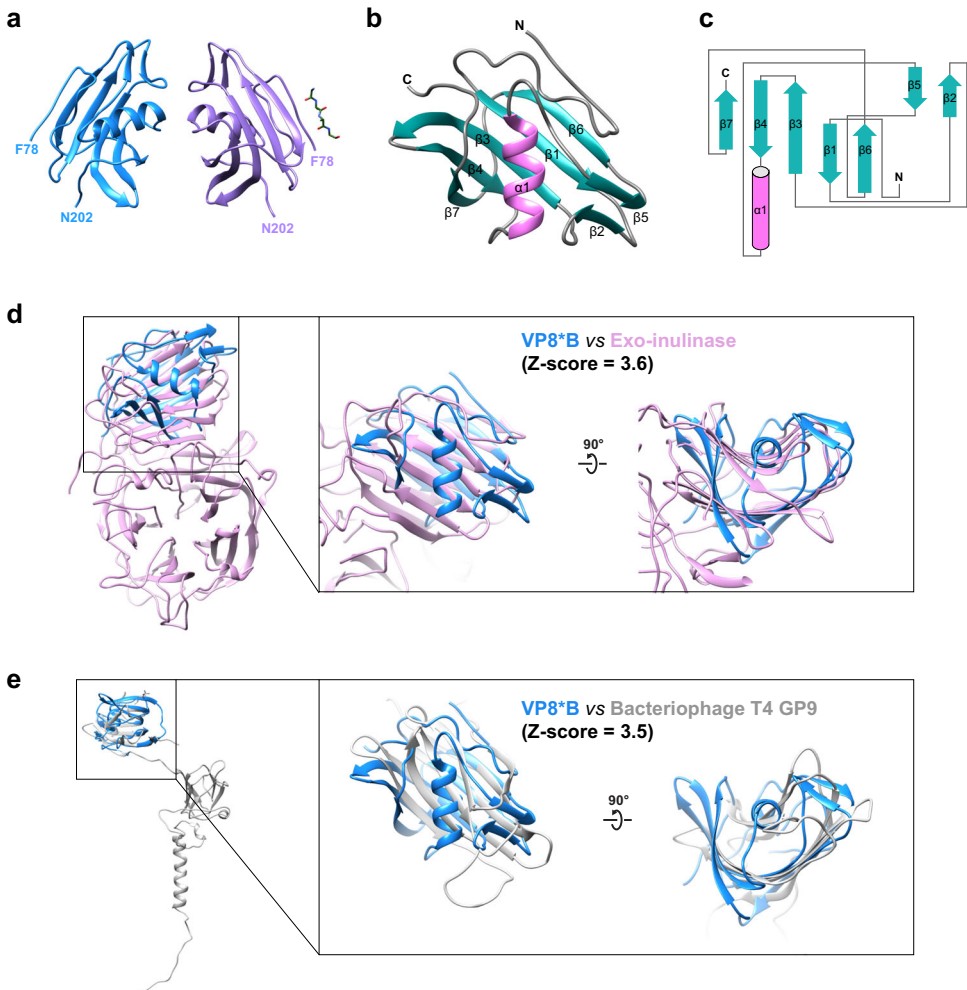

**Fig. 2 Crystal structure VP8*B. a** Crystal structure of VP8*B with chain A and B colored blue and purple, respectively. The co-crystallized peptide is shown as a stick model. **b** Monomer of VP8*B with β-strands and α-helix colored in sea green and pink, respectively, showing the β-sheet wrapped around the helix. **c** The 2D topology diagram of VP8*B. The β-strands of the helix are labeled in (**b**) and (**c**). **d**, **e** Structural comparison of VP8*B with exo-inulinase and bacteriophage T4 gene product 9. The Dali Z-scores are shown in the figures.

barrel domain, whereas the residues that constitute the foot region, composed of residues (491–776), have α-helices as well as β-strands. The trimeric organization of the VP4 spike in RVA is unique in that the central body of the VP4 spike is formed by dimeric interaction between the β-barrel regions of the two subunits. The β-barrel domain of the other subunit, in which the VP8* domain is flexible and not observed in the spike structure, lies between the central dimeric part and proximal foot region. All three subunits of the VP5* contribute to the foot region and interact with three N-terminal α helices of VP8*. The flexible segments of the N-terminus of VP4A run between the two β-barrel domains and connect the N-terminal α helices and the lectin domain.

We used the full-length sequence of the VP4B to predict the structure of VP5*B using AlphaFold2 (Supplementary Fig. 3c). In this prediction, the VP8*B has the same novel fold as predicted by using the VP8*B sequence alone. Evolutionary conservation analysis using the ConSurf server[39] shows that VP8*B is the most variable domain within VP4 (Supplementary Fig. 3d). The residues (214–465) of the VP5*B fold into a β-barrel structure with the same directionality and the disposition of the β-strands as observed in the VP5*A except for the loop regions. The predicted β-barrel of VP5*B matches that of VP5*A with an RMSD of 3.4 Å (Supplementary Fig. 3f). However, the foot

domain of VP5*B, despite having a similar distribution of α-helices and β-strands, superimposed with the corresponding region in VP5*A with a high RMSD of 20.1 Å (Supplementary Fig. 3g), indicating the potential limitations of AlphaFold2 in the ab initio modeling of oligomeric multidomain proteins. Recent cryo-EM studies showed that VP4A can rearrange from an 'upright' to a 'reversed' conformation where the VP5* foot domain is embedded in the membrane[17], suggesting that the structure prediction of the VP5* foot domain in the 'reversed' confirmation should also consider the physical and chemical characteristics of the membrane environment.

**VP8*B binds to glycans containing an N-acetyllactosamine.** To investigate if VP8*B, despite its novel fold, binds to glycans and exhibits any specificity as its counterpart in RVA and RVC, both of which bind to HBGA, we performed high-throughput glycan array screening with the recombinant GST-tagged VP8*B protein[21–24,29]. The glycan array screening shows that VP8*B specifically recognizes glycans containing an N-acetyllactosamine (LacNAc) motif (Fig. 4a, b and Supplementary Table 2), a precursor disaccharide for HBGA synthesis and a universal component of N- and O-glycans and glycolipids[40]. The VP8* of human-bovine reassortant neonate-specific human P[11] RVA

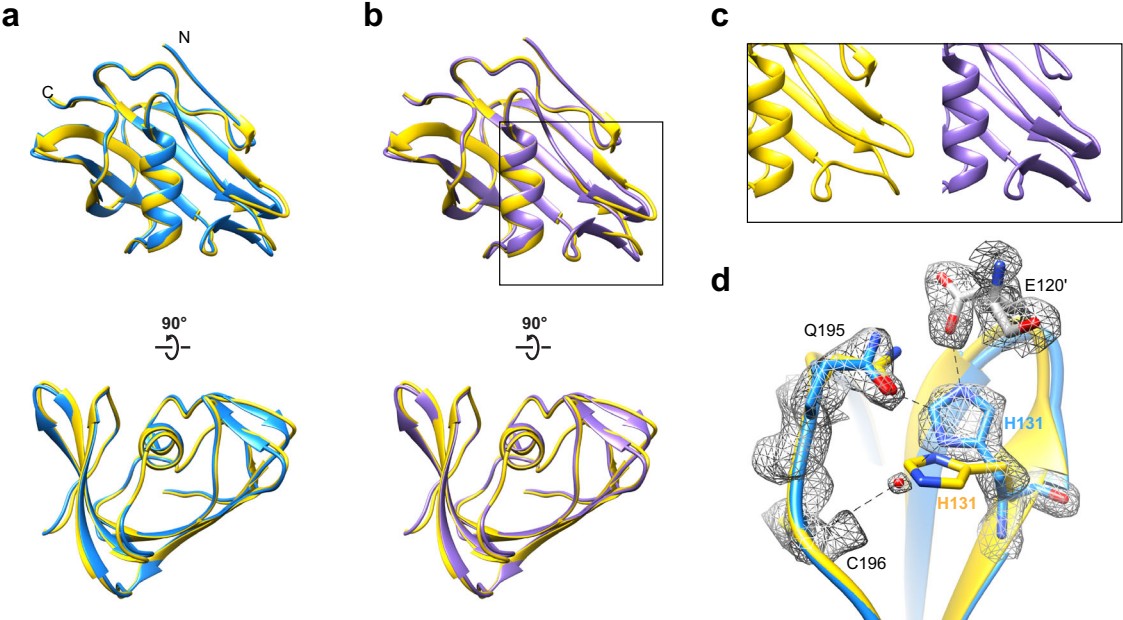

**Fig. 3 Structural comparison of VP8\*B crystal structure with AlphaFold2 prediction. a, b** Superimposition of chain A and B of VP8\*B in the crystallographic asymmetric unit with the AlphaFold2-predicted model. Crystal structure of RVP8\*B with chains A and B colored in blue and purple, respectively. The resides 72–202 of the highest-ranked AlphaFold model is shown as a yellow ribbon diagram. The N- and C-termini of VP8\*B are labeled. **c** The close-up view of the region in the black box in (**b**). The superimposed models are shown side-by-side for clarity. **d** Example of structural difference between the AlphaFold2 model (yellow) and crystal structure (chain A, blue). The hydrogen bonds are shown as black dashed lines. The experimental 2Fo-Fc map of the residues is shown as gray mesh.

and the bovine P[11] RVA also recognize poly-LacNAc glycans[22,31] for cell attachment, suggesting the possibility that LacNAc could also be a cell attachment factor for RVB.

To understand the molecular interactions between VP8\*B and LacNAc disaccharide, we performed molecular docking using AutoDock Vina[41]. The docking pose with the lowest binding free energy of −5.7 kcal/mol shows that the common precursor motif of the selected glycans binds to a shallow pocket on VP8\*B (Fig. 4c and Supplementary Fig. 4). Eight VP8\*B residues are involved in a network of interactions with LacNAc. For example, K153, Y157, T160, S193, and Q195 form hydrogen bonds with LacNAc. Interestingly, the side chain of Q195 also interacts with His131 that is present in different conformations in the predicted and experimental models, suggesting that subtle changes in solution may affect glycan-binding during virus infection.

## Discussion

Although it is common that under evolutionary pressures, a functionally homologous protein within the same genus in a virus family accrues mutations within a conserved polypeptide fold, it rarely evolves to adopt an entirely different fold. Such is the case with the VP8\* domain of RVB, which adopts a new fold without any similarities to the galectin-like fold conserved in the VP8\* of RVA and RVC. All these viruses infect various mammalian species, including humans causing severe diarrheal outbreaks. However, the difference is that RVA and RVC infect predominantly children under the age of 5, whereas RVB, also known as Adult Diarrhea Rotavirus (ADRV), infects predominantly adult populations. It is unclear whether this distinction alone explains such a unique fold of VP8\*B. It likely represents a parallel evolution from a different ancestral animal origin from that of RVA and RVC. AlphaFold2 shows that this novel fold is conserved in other strains of RVB as well. Despite the novel fold, based on our glycan screening study, it is likely

that VP8\*B, similar to the VP8\* of RVA and RVC, retains the same functionality of mediating the initial cell attachment by recognizing specific glycans in the gut. Although LacNAc is one of the common glycans in the human and animal gut, the significance of VP8\*B specificity to this glycan needs further infectivity-based studies, which are currently difficult as the RVB is not yet conducive to cell culture.

Another important aspect of our studies that should be underscored is the accuracy with which AlphaFold2 predicted the fold of VP8\*B, which provided MR search model for our structure determination. In addition to the overall fold, AlphaFold2 also predicted the sidechain orientation with high accuracy when compared with the experimental structure. As exemplified in our studies, such accurate ab initio prediction will be an asset to structural biologists by providing partial or full MR search models for rapidly determining the structures of those proteins without suitable homologous structures. In addition to the VP8\*B structure, AlphaFold2 predicted that VP5\*B has a similar β-barrel fold as VP5\*A, indicating possible conservation of this region. However, the foot region was not as well predicted by AlphaFold2, which as in the RVA may be influenced by the trimeric nature and potential interactions with the outer layer proteins in RVB. Further validation of the VP5\* region of the VP4 spike in RVB requires a cryo-EM structure determination which has to await a successful cell culture adaptation of RVB.

## Methods

**ab initio modeling of RVB VP8\* with AlphaFold2.** The full AlphaFold v2.0 (AlphaFold2) pipeline was obtained from DeepMind and installed on a local workstation[36]. The VP8\*B amino acid sequence (residues 65–223) of human group B rotavirus strain NIV-094456 isolated from gastroenteritis outbreaks in India in 2009 was used as the template[32]. The VP8\*B structure was predicted using the default setting with "–max_template_date=2020-05-14" and "–preset=casp14" which runs with all genetic databases and with eight ensemblings. The AlphaFold2 models of murine, bovine, and porcine VP8\*B and human VP4B were predicted using the same protocol.

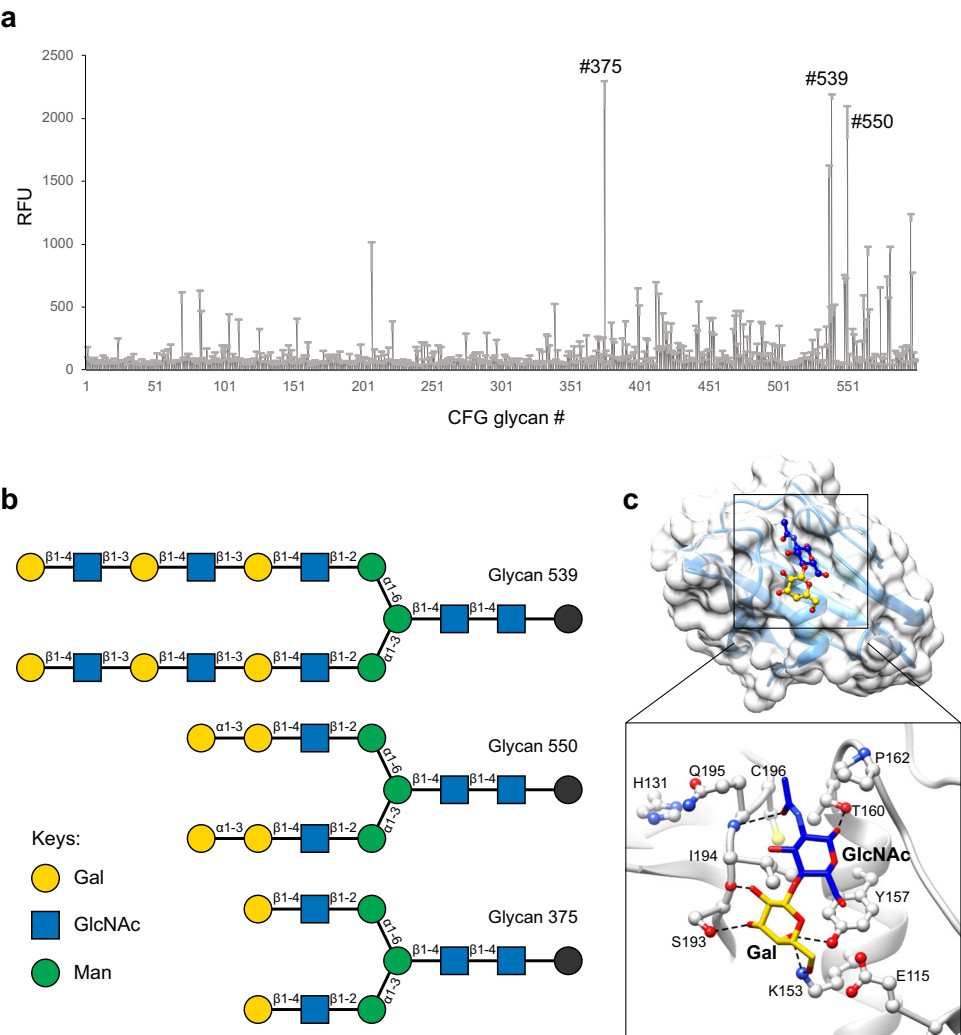

**Fig. 4 Glycan array of VP8*B and molecular docking of the selected glycan. a** The glycan array result of GST-tagged VP8*. The X-axis represents the CFG glycan number, and the Y-axis represents the relative fluorescence units (RFUs) of binding. Standard deviation from the mean of 6 replicates of each glycan printed on the array is provided ($n = 6$). **b** The glycan structures with the highest binding signals were drawn using GlycoGlyph[49]. **c** Molecular docking of the LacNAc disaccharide (Gal-GlcNAc) onto VP8*B. The highest-ranked docked glycan is shown in the stick model, and the protein is shown with a transparent surface. The inset shows how the glycan interacts with VP8*B residues. The glycan and the VP8*B residues are shown in stick and ball-and-stick models, respectively.

**Expression, purification, and crystallization of RVB VP8*.** VP8*B (residues 65–233) of human rotavirus group B (GenBank: AET79992.1) was cloned into expression vectors pQE60 (Qiagen) with a C-terminal His tag and pGEX-4T-1 (Cytiva) with an N-terminal GST-tag. Recombinant C-terminal His-tagged VP8*B was expressed in *E. coli* BL21(DE3) cells (Novagen) by inducing cells with 200 μM IPTG at 25 °C for 16 h. Cells were mechanically lysed and purified with Ni-NTA resin by batch purification. The His-tagged VP8*B was further purified by size exclusion column Superdex75 (GE healthcare) with 10 mM Tris, pH 8.0, 100 mM NaCl at 4 °C. The concentration of the purified protein was determined by measuring the absorbance at 280 nm and using an absorption coefficient of 12,950 $M^{-1}$ $cm^{-1}$ for VP8*B calculated using ProtPraram on the ExPASy server[42].

Recombinant N-terminal GST-tagged VP8*B was expressed in *E. coli* BL21(DE3) cells (Novagen) by inducing cells with 200 μM IPTG at 25 °C for 16 h. Cells were mechanically lysed and purified with Glutathione Sepharose resin by batch purification (Thermo Scientific). The purified elution was then dialyzed into 10 mM Tris pH 8.0, 100 mM NaCl overnight. The GST-VP8* was further purified by size exclusion column Superdex200 (GE healthcare) with 10 mM Tris pH 8.0, 100 mM NaCl at 4 °C. The concentration of the purified protein was determined by measuring the absorbance at 280 nm and using an absorption coefficient of 56,185 $M^{-1}$ $cm^{-1}$ for GST-VP8*B calculated using ProtPraram on the ExPASy server[42].

Crystallization screenings for VP8*B at the concentration of 20 mg/ml were carried out by hanging-drop vapor diffusion using the Mosquito crystallization robot (TTP LabTech) at 20 °C. After 80 days, VP8*B was crystallized under the

condition with 0.2 M ammonium sulfate, 30% PEG 2 K MME, 0.1 M Na acetate, pH 4.6. Crystals were flash-frozen directly in liquid nitrogen.

**Data collection and structure determination.** X-ray diffraction data for the VP8*B crystals were collected on a PILATUS detector of beamline 5.0.1 at Advanced Light Source (Berkeley, CA). Diffraction data were processed using HKL2000[43]. The flexible loops at the N- and C-termini of the highest-ranked AlphaFold2 model were removed, and only the residues 83–192 were used as the search model for MR using Phaser[44]. A single solution with two molecules of VP8*B in the asymmetric unit was found by Phaser. The automated model building with the MR solution was carried out using ARP/wARP[44]. Iterative cycles of refinement with simulated annealing and manual model building were performed using Phenix[45] and Coot[37]. Data refinement and statistics are shown in Table 1. Figures were prepared using Chimera[46].

**Glycan array screening.** The glycan-binding specificity of VP8*B was investigated on a glycan array comprised of 600 glycans at the Emory Glycomics and Molecular Interactions Core (EGMIC), Emory University[22–24]. Recombinant GST-tagged VP8*B at 5 μg/ml concentration in binding buffer (20 mM Tris-HCl pH 7.4, 150 mM sodium chloride, 2 mM calcium chloride, 2 mM magnesium chloride, 0.05% Tween 20, 1% BSA) was applied to the glycan array, and bound protein was detected using 5 μg/ml Alexa Flour 647 anti-GST tag monoclonal antibody (8–326) (Thermo Fisher Scientific, cat# MA4-004-A647). A summary of the glycan array

result is given in Supplementary Table 2. The complete list of glycans is provided in Supplementary Data 1.

**Molecular docking with AutoDock Vina**. The LacNAc disaccharide (Gal-GlcNAc) was docked onto VP8*B using AutoDock Vina[41]. The protein was processed by adding polar hydrogen atoms using AutoDockTools[47]. VP8*B was treated as a rigid body, while the glycan was allowed to have all possible rotational angles. The grid box was centered on VP8*B with the size $(40 \times 40 \times 40$ Å) of the box adjusted to cover the entire protein. Docking was carried out with an exhaustiveness of 24. The pose with the lowest binding free energy of $-5.7$ kcal/mol was viewed using ViewDock in Chimera, and the molecular interactions were analyzed using LigPlot+ (v2.2.4)[48].

**Evolutionary conservation analysis**. Evolutionary conservation was analyzed using the ConSurf web server[39] using the full-length ADRV VP4B AlphaFold model as a query with default parameters. Multiple sequence alignment was built using MAFFT. The homologs were collected from non-redundant (NR) sequences from GenBank CDS translations with 35–95% identity. The calculation was performed with 30 unique sequences using the Bayesian calculation method.

**Statistics and reproducibility**. For the glycan array results, standard deviation and % coefficient of variation (%CV) are calculated from the mean of 6 replicates ($n = 6$) of each glycan printed on the array and are from one of two independent experiments with similar results.

**Reporting summary**. Further information on research design is available in the Nature Research Reporting Summary linked to this article.

## Data availability

Atomic coordinates and structure factors for the crystal structure of VP8*B have been deposited in the Protein Data Bank under the accession code 7RSW. The authors declare that all other data supporting the findings of this study are available within the paper and its supplementary information files. Source data are provided with this paper.

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

## Acknowledgements
We acknowledge the support from NIH grants AI36040 (to B.V.V.P.), AI080656 and P30 DK56338 (to M.K.E. and S.E.C.), and the Robert Welch Foundation (Q1279) to B.V.V.P. W.S. was supported through the training fellowship from the Gulf Coast Consortia, on the Training Interdisciplinary Pharmacology Scientists (TIPS) Program (Grant No. T32 GM120011). This research used the Advanced Light Source resources, a DOE Office of Science User Facility under contract no. DE-AC02-05CH11231. The ALS-ENABLE beamlines are supported by the National Institutes of Health, National Institute of General Medical Sciences, grant P30 GM124169-01.

## Author contributions
L.H. and B.V.V.P. designed the research. L.H. predicted the AlphaFold2 model, determined the VP8*B crystal structure, and analyzed the data. W.S. performed the protein purification and crystallization, and B.S. performed X-ray diffraction data collection. Y.L. and D.F.S. contributed to glycan array experiments and analysis. S.E.C. and M.K.E. provided advice on the result analyses. L.H. and B.V.V.P. wrote the manuscript. All authors reviewed, edited, and approved the final manuscript.

## Competing interests
The authors declare no competing interests.
