## [Peer Review File · Communications Biology]

Reviewers' comments:

Reviewer #1 (Remarks to the Author):

This paper illustrated the structure of VP8* of group B rotavirus (VP8*B), which is quite different to the VP8*s of group A and C rotaviruses. They used the developed AlphaFold2 to generate the model and successfully determined the crystal structure of human VP8*B. VP4B structure model was also predicted by AlphaFold2 and compared with VP4A. The glycan array screening of VP8*B was performed and showed that VP8*B specifically recognized glycans containing an N-acetylglucosamine (GlcNAc) motif. The structural and functional characterization of VP8*B provides more understanding of different RV groups. The work is convincing and well performed. For the glycan binding specificities, it could be better if the authors can provide more evidence by different methods such as ELISA or BLI interactions.

Reviewer #2 (Remarks to the Author):

The manuscript by Hu et al describes determination by x-ray crystallography of the structure of the globular part of rotavirus B (RVB) VP8*, likely to be the receptor-binding module. Absence of sensible amino-acid sequence alignment with the sequences of RVA and RVCVP8* galectin-like domains, and failure of molecular replacement efforts using those structure suggested a different structure for RVB VP8*. The authors used an AlphaFold2 prediction as the MR model instead, and obtained a 1.8Å resolution structure. The domain does not resemble any known folded globular domain currently in the PDB. Glycan array analysis showed that it binds oligo- and polysaccharides that contain N-acetylglucosamine (the Gal-GlcNAc disaccharide).

The MS is straightforward, interesting, and (like all good structural biology papers) convincing. It should be published with only modest modifications to the text, as suggested below.

First, two general points

1. The authors place too much interest on use of AlphaFold2, when the real interest is the RVB receptor-binding domain. Although use of AlphaFold2 models for MR is obviously very new, that application of its powerful predictive capabilities is so obvious (and probably already shown by published work) that including it in the title seems to this reviewer to distract from the more interesting results.

2. As I understand the (ugly and vague, but generally used) term, "novel fold" simply means a domain structure different from any currently in the PDB. By that definition, the structure here is definitely a "novel fold". But that does NOT mean that evolution has "sculpted" it (another hideous and here inappropriate word) from scratch on VP8*. VP4 is modular, and what the authors have shown very nicely is that one of those modules in RVB VP4 has a different origin than do the corresponding modules in RVA and RVC VP4s. But there is no reason to suppose that it evolved "in place". This reviewer finds it more likely, given everything we know about multi-modular proteins, that it simply recombined into VP4 from some other source. Just because that source is not (yet) in the PDB is irrelevant to the argument -- there will be lots more "novel folds" before the catalog of all folded structures is complete. Although there's no evidence now about what the N-terminal 25 or so residues of RVB VP4 are doing, should a structure show that they form an α helix like the one in RVA VP4, cementing the three foot-modules together, then I would probably place bets on the insertion-from-elsewhere model. Since can't yet recapitulate in the lab the evolution of structures like this, everyone is entitled to their guess, of course, but it would be wise here to be more balanced.

Specific suggestions, some of which are purely to improve clarity and style, while others have scientific substance:

line 4: ... has a novel fold, distinct from that of the conserved, galectin-like fold ... (you can't be "novel compared to" any more than you can be "unique compared to"; since there's no difference

between "highly conserved" and "conserved", for the same reason, get rid of "highly", which is just a meaningless intensifier)

line 6: ... VP8*B has little sequence similarity with ... (in fact, it has none -- the only way the alignment gave anything at all in Fig. S1a is by insertion of clearly unphysical gaps in the A and C sequences, so the "11%" is meaningless)

line 16: see general point 2, above. I'd argue that the correct sentence here should be "Our study illustrates how evolution can incorporate structurally distinct folds with similar functionality within homologous proteins in the same virus family."

line 24: move the phrase ", and to a lesser extent group C rotaviruses (RVC)," so that it intervenes between "(RVA)" and "are the causative agents"

line 43: ... have shown that the galectin-like VP8* domain of human RVA and RVC recognize various cellular glycans, depending on genotype (Figs. 1a-b)(2-7,27,28).

line 46: delete redundant "which are the determinants of blood groups." Anyone who reads this will know that what a histo-blood group antigen is.

lines 49-51. See comment on line 6, above. The alignment enforces ridiculous insertions into RVA and RVC VP8*, so even the 11% is clearly an artifact of the particular gap penalty imposed, etc. Anyone who has thought about it would have looked at that alignment and said "this is not the same structure, and I'll tell my student/postdoc they're wasting their time trying any of the RVA or RBC structures as MR, and they'll get the wrong answer if they seem to get a possible one".

line 52: Get rid of the "given such low sequence similarity" clause, for the reason just stated.

line 60: "was unsuccessful"

line 64: delete "highly" (see above)

line 65: In the AlphaFold2 model, residues 83-192 fold into several beta-strands that form a twisted sheet clasping ...

line 99: delete "successfully" -- "having determined" implies success, and you'd never have written "having unsuccessfully determined ..."

line 102: Are those alpha-carbon RMSDs or all-atom RMSDs?

line 120-125. The description of the VP8*-VP5* trimer is partly inaccurate. First, the third VP8* has almost certainly been removed by trypsin at residue 26 -- the very short N-terminal fragment that remains doesn't show up on SDS-PAGE, but it has to be there, as there are three N-terminal alpha helices (not two as something in the paragraph implies). "Loops" is a poor descriptor for the segments between those helices and the lectin domain. They are segments that extend from the helices at the "bottom" to the N-termini of the lectin domains at the "top", and they run between the two beta-barrel domains, so "skirting" them, which would imply to a reader running along the outside, gives an incorrect verbal image.

line 133: The principal reason that AlphaFold fails with the foot is probably that it has two very different structures, one in the upright conformation and some other, embedded in the membrane, on the reversed conformation. Moreover, unless the N-terminal helix were part of the fold attempt (with an arbitrary linker to about residue 500 in the RRV sequence), one wouldn't expect any program to get the foot right.

line 145: Did you try to get a crystal structure of the complex with Gal-GlcNac or some other short glycan containing that motif?

Stephen Harrison (who apologizes to the authors for an unconscionable delay in reviewing this MS)

Reviewer #3 (Remarks to the Author):

Summary:

Rotavirus continues to be an important cause of gastroenteritis in humans. Rotavirus basic science research has primarily focused on group A family members. Little is understood about the structure and entry mechanisms of divergent rotavirus groups. Here, Hu, et al. expand our understanding of the ligand binding mechanism of Group B rotaviruses by characterizing and determining the crystal structure of a Group B rotavirus VP8*, the predicted lectin binding domain of the Group B rotavirus VP4 entry protein. Phasing for the crystal structure came by using the VP8* structural model predicted by AlphaFold2, an AI program that predicts protein folds. Group B rotavirus VP8* has a strikingly different fold than their group A relatives. Using a glycan binding assay, the authors determine that this VP8* preferentially binds specific ligands. They also use modeling programs to predict the sugar binding pocket in VP8*. This work shows how rotavirus family members have acquired or adapted protein domains to serve similar functions in glycan binding for cellular entry, as well as the utility of ab initio protein structure prediction to provide a starting point for experimental structures.

This manuscript provides a structural basis how Group B rotaviruses can bind to cells and their in vitro preferences for surface glycans. These findings will have impact in understanding host and tissue tropism, viral evolution, and will provide insight into how neutralizing antibodies block Group B rotavirus entry. As such, this work will be of interest to virologists studying rotavirus, virus evolution, and vaccine design. The manuscript was well written and, for the most part, adequately cited and reported appropriate crystallography statistics. For the most part, the methods were described in enough detail to repeat the experiments described. Significant emphasis was placed on the use of AlphaFold2 for phasing. While this work is an example an applied use and accuracy of AlphaFold2, the impact determined for AlphaFold2 applications were modest. Experimental phasing for the crystal structure was not attempted and would most likely have led to a similar conclusion. While this emphasis was not distracting, it did not greatly affect the manuscript impact. Conclusions made about the glycan preference and binding footprint were intriguing, but additional experiments would greatly support the validity of the proposed glycan binding site.

Scott Aoki

Major:

1. The CC1/2 should be reported.
2. Please explain how you came to the resolution cutoff decision in the methods. 61% completeness in the last shell is very modest.
3. The four-peptide chain should be described in more detail. Did the sequence match a region in the VP8* N or C terminus? How strong is the density? The current text implies that the authors propose this peptide is an artifact of crystallography versus biological relevance. If so, this idea should be stated explicitly.
4. Sequence conservation overlaid on the determined structure (e.g. Protskin) between Rotavirus B (RVB)s will help strengthen the argument that the VP8* fold identified is conserved across all RVBs.
5. The proposed glycan binding site appears plausible, but the manuscript will be strengthened greatly with additional binding/mutagenesis studies confirming that this identified pocket is indeed the location of glycan binding observed in vitro.

Minor:

1. I assume this will be added by the final revisions, but subheaders for Introduction, Results, and Discussion (or Results and Discussion) will be helpful.
2. The title implies that the structural work was all done by AlphaFold2, when in reality the manuscript provides experimental evidence confirming the proposed structure. Recommend title

revision to, "Novel fold of rotavirus glycan-binding domain predicted by AlphaFold2 and determined by X-ray crystallography," or something of this nature to not undersell the authors' efforts.

3. Lines 31-35, please cite reference(s) connecting new SARS variants to severe disease.

4. Lines 37-39, please reference earlier cryoEM work that showed what is described in the text.

5. Line 44, please consider adding (Xu, et al. PNAS 2021, DOI: 10.1073/pnas.2107963118)

6. Fig 1: It will be helpful to have a Group B rotavirus linear diagram of VP4 and the VP8* region investigated, similar to Fig S2a.

7. Line 77: describe or cite how you calculated RMSD.

8. Line 76-77: Can you clarify which models (crystal structure asu copies, AlphaFold2 model, etc) are being compared in lines 76-77?

9. Any effort to limit phase bias should be described in the methods.

10. Line 136: please cite references for binding to HBGA, particularly to refer to RVC binding that was not covered in the introduction.

11. Line 136-137: Please site glycan array reference.

12. Line 175: Remove "reasonable accuracy." This statement requires experimental evidence that VP5*B has a similar fold to VP5*A.

13. Line 176: Remove "understandably," since we do not understand (only speculate) the reasons for poor modeling.

14. Line 183: please cite AlphaFold2.

15. Lines 196, 203: Remove comma in "10 mM Tris, pH 8.0..."

16. Was any mass spec performed on the protein in the crystals? If so, please mention in at least the methods.

17. Line 221, glycan array screening: Wasn't this the same/similar array screening previously used? If so, please cite.

Communications Biology manuscript COMMSBIO-21-2453-T

We thank all the reviewers for their positive comments and suggestion. We have addressed each of the reviewers' comments in detail and revised the text accordingly. Our responses are in blue.

Responses to Reviewer #1 comments:

This paper illustrated the structure of VP8* of group B rotavirus (VP8*B), which is quite different to the VP8*s of group A and C rotaviruses. They used the developed AlphaFold2 to generate the model and successfully determined the crystal structure of human VP8*B. VP4B structure model was also predicted by AlphaFold2 and compared with VP4A. The glycan array screening of VP8*B was performed and showed that VP8*B specifically recognized glycans containing an N-acetyllactosamine (LacNAc) motif. The structural and functional characterization of VP8*B provides more understanding of different RV groups. The work is convincing and well performed.

Response: We thank the reviewer for the positive comments.

For the glycan binding specificities, it could be better if the authors can provide more evidence by different methods such as ELISA or BLI interactions.

Response: We thank the reviewer for the suggestion. Based on the glycan array, we used commercially available simple monovalent glycans such as LacNAc disaccharide and LNnT tetrasaccharide and tested their binding using BLI, ELISA, and STD NMR, but did not see any binding. It is likely that the binding of the multivalent glycans we observed using the glycan array (Figure 4) have much higher avidity towards VP8*B. Unfortunately, these branched glycans, especially the ones with long Gal-GlcNAc chains, are not commercially available and require custom synthesis. However, the glycan array data provide clear evidence for the ability of VP8*B to recognize glycans.

Responses to Reviewer #2 comments:

The manuscript by Hu et al describes determination by x-ray crystallography of the structure of the globular part of rotavirus B (RVB) VP8*, likely to be the receptor-binding module. Absence of sensible amino-acid sequence alignment with the sequences of RVA and RVCVP8* galectin-like domains, and failure of molecular replacement efforts using those structure suggested a different structure for RVB VP8*. The authors used an Alphafold2 prediction as the MR model instead, and obtained a 1.8Å resolution structure. The domain does not resemble any known folded globular domain currently in the PDB. Glycan array analysis showed that it binds oligo- and polysaccharides that contain N-acetyllactosamine (the Gal-GlcNAc disaccharide).

The MS is straightforward, interesting, and (like all good structural biology papers) convincing. It should be published with only modest modifications to the text, as suggested below.

Response: We thank the reviewer for the positive comments.

First, two general points

1. The authors place too much interest on use of AlphaFold2, when the real interest is the RVB receptor-binding domain. Although use of AlphaFold2 models for MR is obviously very new, that application of its powerful predictive capabilities is so obvious (and probably already shown by published work) that including it in the title seems to this reviewer to distract from the more interesting results.

Response: We thank the reviewer for the suggestion and agree with the suggestion. We have changed the title to 'Novel fold of rotavirus glycan-binding domain predicted by AlphaFold2 and determined by X-ray crystallography'.

2. As I understand the (ugly and vague, but generally used) term, "novel fold" simply means a domain structure different from any currently in the PDB. By that definition, the structure here is definitely a "novel fold". But that does NOT mean that evolution has "sculpted" it (another hideous and here inappropriate word) from scratch on VP8*. VP4 is modular, and what the authors have shown very nicely is that one of those modules in RVB VP4 has a different origin than do the corresponding modules in RVA and RVC VP4s. But there is no reason to suppose that it evolved "in place". This reviewer finds it more likely, given everything we know about multi-modular proteins, that it simply recombined into VP4 from some other source. Just because that source is not (yet) in the PDB is irrelevant to the argument -- there will be lots more "novel folds" before the catalog of all folded structures is complete. Although there's no evidence now about what the N-terminal 25 or so residues of RVB VP4 are doing, should a structure show that they form an a helix like the one in RVA VP4, cementing the three foot-modules together, then I would probably place bets on the insertion-from-elsewhere model. Since can't yet recapitulate in the lab the evolution of structures like this, everyone is entitled to their guess, of course, but it would be wise here to be more balanced.

Response: Excellent point. We have now revised the entire abstract taking into consideration the overall tone of the comments made by the reviewer and shortened it to ~150 words as per the journal requirement (Page 2, lines 2-12).

Specific suggestions, some of which are purely to improve clarity and style, while others have scientific substance:

Response: We have edited the manuscript in a point-by-point manner below.

line 4: ... has a novel fold, distinct from that of the conserved, galectin-like fold ... (you can't be "novel compared to" any more than you can be "unique compared to"; since there's no difference between "highly conserved" and "conserved", for the same reason, get rid of "highly", which is just a meaningless intensifier)

Response: We have deleted "highly" in the revised abstract.

line 6: ... VP8*B has little sequence similarity with ... (in fact, it has none -- the only way the alignment gave anything at all in Fig. S1a is by insertion of clearly unphysical gaps in the A and C sequences, so the "11%" is meaningless)

Response: We have changed "low sequence similarity" to "lack of sequence similarity" in the revised abstract (Page 2, line 5).

line 16: see general point 2, above. I'd argue that the correct sentence here should be "Our study illustrates how evolution can incorporate structurally distinct folds with similar functionality within homologous proteins in the same virus family."

Response: We have changed this sentence to " Our study illustrates how evolution can incorporate structurally distinct folds with similar functionality in a homologous protein within the same virus genus" (Page 2, lines 11-12)

line 24: move the phrase ", and to a lesser extent group C rotaviruses (RVC)," so that it intervenes between "(RVA)" and "are the causative agents"

Response: We have changed this sentence to "While the group A rotaviruses (RVA) and to a lesser extent group C rotaviruses (RVC) are the causative agents of most gastroenteric infections worldwide, the group B rotaviruses (RVB) have been associated with large epidemic outbreaks of severe gastroenteritis in China." (Page 2, line 20)

line 43: ... have shown that the galectin-like VP8* domain of human RVA and RVC recognize various cellular glycans, depending on genotype (Figs. 1a-b)(2-7,27,28).

Response: We have changed this sentence to "Extensive structural studies have shown that the galectin-like VP8* of human RVA and RVC recognize various cellular glycans in a genotype-dependent manner." (Page 3, lines 39-41)

line 46: delete redundant "which are the determinants of blood groups." Anyone who reads this will know that what a histo-blood group antigen is.

Response: We have deleted "which are the determinants of blood groups." (Page 3, line 43)

lines 49-51. See comment on line 6, above. The alignment enforces ridiculous insertions into RVA and RVC VP8*, so even the 11% is clearly an artifact of the particular gap penalty imposed, etc. Anyone who has thought about it would have looked at that alignment and said "this is not the same structure, and I'll tell my student/postdoc they're wasting their time trying any of the RVA or RBC structures as MR, and they'll get the wrong answer if they seem to get a possible one".

Response: We have changed this sentence to "The VP8*B shares no sequence identity with either VP8*A or VP8*C (Figure S1a and Table S1)" (Page 3, line 47)

line 52: Get rid of the "given such low sequence similarity" clause, for the reason just stated.

Response: We have removed the clause "given such low sequence similarity." (Page 4, lines 57-58)

line 60: "was unsuccessful"

Response: "not successful" changed to "unsuccessful". (Page 4, line 65)

line 64: delete "highly" (see above)

Response: We have deleted "highly".(Page 4, line 69)

line 65: In the AlphaFold2 model, residues 83-192 fold into several beta-strands that form a twisted sheet clasping ...

Response: We have changed "are well folded into" to "fold into". (Page 4, line 70)

line 99: delete "successfully" -- "having determined" implies success, and you'd never have written "having unsuccessfully determined ..."

Response: "successfully" deleted. (Page 6, line 110)

line 102: Are those alpha-carbon RMSDs or all-atom RMSDs?

Response: Those are alpha-carbon RMSDs. We have edited the sentence to clarify this. (Page 6, lines 112-113)

line 120-125. The description of the VP8*-VP5* trimer is partly inaccurate. First, the third VP8* has almost certainly been removed by trypsin at residue 26 -- the very short N-terminal fragment that remains doesn't show up on SDS-PAGE, but it has to be there, as there are three N-terminal alpha helices (not two as something in the paragraph implies). "Loops" is a poor descriptor for the segments between those helices and the lectin domain. They are segments that extend from the helices at the "bottom" to the N-termini of the lectin domains at the "top", and they run between the two beta-barrel domains, so "skirting" them, which would imply to a reader running along the outside, gives an incorrect verbal image.

Response: We have edited this paragraph according to the reviewer's suggestion. (Page 7, lines 134-137)

line 133: The principal reason that AlphaFold fails with the foot is probably that it has two very different structures, one in the upright conformation and some other, embedded in the membrane, on the reversed conformation. Moreover, unless the N-terminal helix were part of the fold attempt (with an arbitrary linker to about residue 500 in the RRV sequence), one wouldn't expect any program to get the foot right.

Response: We have added the discussion of two VP4 conformations in the revised manuscript. (Page 8, lines 147-151)

line 145: Did you try to get a crystal structure of the complex with Gal-GlcNAc or some other short glycan containing that motif?

Response: We have tried to co-crystallize the simple LacNAc disaccharide moiety with VP8*B and obtained some crystals for structure determination. However, there was no electron density that account for the glycan in the crystals. It might be due to the low affinity of the disaccharide to VP8*B or the crystallization condition did not favor the complex formation. Perhaps more complex branched glycans as shown in the glycan array will have to be explored.

Responses to Reviewer #3 comments:

Summary:

Rotavirus continues to be an important cause of gastroenteritis in humans. Rotavirus basic science research has primarily focused on group A family members. Little is understood about the structure and entry mechanisms of divergent rotavirus groups. Here, Hu, et al. expand our understanding of the ligand binding mechanism of Group B rotaviruses by characterizing and determining the crystal structure of a Group B rotavirus VP8*, the predicted lectin binding domain of the Group B rotavirus VP4 entry protein. Phasing for the crystal structure came by using the VP8* structural model predicted by AlphaFold2, an AI program that predicts protein folds. Group B rotavirus VP8* has a strikingly different fold than their group A relatives. Using a glycan binding assay, the authors determine that this VP8* preferentially binds specific ligands. They also use modeling programs to predict the sugar binding pocket in VP8*. This work shows how rotavirus family members have acquired or adapted protein domains to serve similar functions in glycan binding for cellular entry, as well as the utility of ab initio protein structure prediction to provide a starting point for experimental structures.

This manuscript provides a structural basis how Group B rotaviruses can bind to cells and their in vitro preferences for surface glycans. These findings will have impact in understanding host and tissue tropism, viral evolution, and will provide insight into how neutralizing antibodies block Group B rotavirus entry. As such, this work will be of interest to virologists studying rotavirus, virus evolution, and vaccine design. The manuscript was well written and, for the most part, adequately cited and reported appropriate crystallography statistics. For the most part, the methods were described in enough detail to repeat the experiments described. Significant emphasis was placed on the use of AlphaFold2 for phasing. While this work is an example an applied use and accuracy of AlphaFold2, the impact determined for AlphaFold2 applications were modest. Experimental phasing for the crystal structure was not attempted and would most likely have led to a similar conclusion. While this emphasis was not distracting, it did not greatly affect the manuscript impact. Conclusions made about the glycan preference and binding footprint were intriguing, but additional experiments would greatly support the validity of the proposed glycan binding site.

Scott Aoki

Response: We thank the reviewer for the remarks and suggestions. We have edited the manuscript in a point-by-point manner below.

Major:

1. The CC1/2 should be reported.

Response: We have included the CC1/2 in Table S2 in the revised manuscript. (Page 18)

2. Please explain how you came to the resolution cutoff decision in the methods. 61% completeness in the last shell is very modest.

Response: We cut the resolution based on the $I/\sigma I$ is 2.31, which is >2 , and the Rmerge in the highest resolution shell is 26.2%.

3. The four-peptide chain should be described in more detail. Did the sequence match a region in the VP8* N or C terminus? How strong is the density? The current text implies that the authors propose this peptide is an artifact of crystallography versus biological relevance. If so, this idea should be stated explicitly.

Response: We have built amino acids "GAAG" based on the electron density. There is no clear site chain density that allows us to assign any of the VP8* N or C terminus residues. Therefore, we think it will be speculative to propose any biological relevance of the peptide.

We have further stated this and provided a new supplemental figure with the stereo image Fig.S2 of the electron density map that contains this peptide. (Page 27)

4. Sequence conservation overlaid on the determined structure (e.g. Protskin) between Rotavirus B (RVB)s will help strengthen the argument that the VP8* fold identified is conserved across all RVBs.

Response: We have carried out the ConSurf analysis using the VP4B AlphaFold model that shows low conservation in the VP8* region. We have added a new supplemental figure (Fig. S3d) to illustrate this as suggested by the reviewer. (Page 28)

5. The proposed glycan binding site appears plausible, but the manuscript will be strengthened greatly with additional binding/mutagenesis studies confirming that this identified pocket is indeed the location of glycan binding observed in vitro.

Response: We have tried to co-crystallize the simple LacNAc disaccharide moiety with VP8*B and obtained some crystals for structure determination. However, there was no electron density that account for the glycan in crystals. It might be due to the low affinity of the disaccharide to VP8*B or the crystallization condition did not favor the complex formation. Perhaps more complex branched glycans as shown in the glycan array will have to be explored.

Minor:

1. I assume this will be added by the final revisions, but subheaders for Introduction, Results, and Discussion (or Results and Discussion) will be helpful.

Response: We have formatted the entire manuscript including the abstract (~150 words) following the guidelines of Communications Biology.

2. The title implies that the structural work was all done by AlphaFold2, when in reality the manuscript provides experimental evidence confirming the proposed structure. Recommend title revision to, "Novel fold of rotavirus glycan-binding domain predicted by AlphaFold2 and determined by X-ray crystallography," or something of this nature to not undersell the authors' efforts.

Response: We thank the reviewer for the suggestion. Taking into consideration similar comments by Reviewer 2 also, we have changed the title to "Novel fold of rotavirus glycan-binding domain predicted by AlphaFold2 and determined by X-ray crystallography"

3. Lines 31-35, please cite reference(s) connecting new SARS variants to severe disease.

Response: We have cited: Viana, R. *et al.* Rapid epidemic expansion of the SARS-CoV-2 Omicron variant in southern Africa. *Nature*, doi:10.1038/s41586-022-04411-y (2022). (Page 3, line 32)

4. Lines 37-39, please reference earlier cryoEM work that showed what is described in the text.

Response: We have added one earlier cryoEM work: Shaw, A. L. *et al.* Three-dimensional visualization of the rotavirus hemagglutinin structure. *Cell* **74**, 693-701, doi:10.1016/0092-8674(93)90516-s (1993). (Page 3, line 34-36)

5. Line 44, please consider adding (Xu, et al. PNAS 2021, DOI: 10.1073/pnas.2107963118)

Response: We have cited: Xu, S. *et al.* Structural basis of P[II] rotavirus evolution and host ranges under selection of histo-blood group antigens. *Proc Natl Acad Sci U S A* **118**, doi:10.1073/pnas.2107963118 (2021). (Page 3, line 41)

6. Fig 1: It will be helpful to have a Group B rotavirus linear diagram of VP4 and the VP8* region investigated, similar to Fig S2a.

Response: We thank the reviewer for the suggestion. We have added a new figure of the linear diagram of VP4B in Figure S3a to show it side-by-side with VP4A. (Page 28)

7. Line 77: describe or cite how you calculated RMSD.

Response: We have edited the sentence to "These two molecules in the asymmetric unit of the crystal structure superimposed with a C α RMSD of 0.037 Å using Secondary Structure Matching (SSM) superpose in COOT³⁷." and cited the program COOT. (Page 5, lines 82-84)

8. Line 76-77: Can you clarify which models (crystal structure asu copies, AlphaFold2 model, etc) are being compared in lines 76-77?

Response: We have clarified that two crystal structure asu copies were compared and changed the sentence as shown in our response to this reviewer's comment #7. (Page 5, lines 82-84)

9. Any effort to limit phase bias should be described in the methods.

Response: We have used ARP/wARP to rebuild the model and used simulated annealing, which allows the structure to escape local minima, in Phenix to remove phase bias. We have edited the method. (Page 11, line 238)

10. Line 136: please cite references for binding to HBGA, particularly to refer to RVC binding that was not covered in the introduction.

Response: We have added citations here, including "Sun, X. *et al.* Human Group C Rotavirus VP8*s Recognize Type A Histo-Blood Group Antigens as Ligands. *J Virol* **92**, doi:10.1128/JVI.00442-18 (2018)." (Page 8, line 155)

11. Line 136-137: Please site glycan array reference.

Response: We have added the reference (Page 8, line 155)

12. Line 175: Remove "reasonable accuracy." This statement requires experimental evidence that VP5*B has a similar fold to VP5*A.

Response: We have removed "reasonable accuracy." as suggested by the reviewer. (Page 10, line 194)

13. Line 176: Remove "understandably," since we do not understand (only speculate) the reasons for poor modeling.

Response: We have removed "understandably," as suggested by the reviewer. (Page 10, line 195)

14. Line 183: please cite AlphaFold2.

Response: We have added the reference. (Page 10, line 203)

15. Lines 196, 203: Remove comma in "10 mM Tris, pH 8.0..."

Response: We have removed the comma. (Page 11, lines 222 and 224)

16. Was any mass spec performed on the protein in the crystals? If so, please mention in at least the methods.

Response: We did not perform mass spec on the protein in the crystal.

17. Line 221, glycan array screening: Wasn't this the same/similar array screening previously used? If so, please cite.

Response: We have used the same assay for our previous publications. We have now added the citation here as suggested by the reviewer. (Page 12, line 244)

REVIEWERS' COMMENTS:

Reviewer #2 (Remarks to the Author):

The authors have responded effectively and carefully to comments of the several reviewers. I recommend prompt publication of this interesting contribution to virology generally and to rotavirus biology in particular.

Stephen Harrison

Reviewer #3 (Remarks to the Author):

This manuscript by Hu, et al. provides a structural basis how Group B rotaviruses can bind to cells and their in vitro preferences for surface glycans. In the revision, the authors revised or addressed all of my (reviewer #3) concerns. The work is novel and will be of interest to virologists and structural biologists.

A very minor typo:

Line 222: add space between 10 and mM in "10mM Tris pH 8.0"